

# If you build it, they will come: rapid colonization by dragonflies in a new effluent-dependent river reach

Michael T. Bogan[1], Drew Eppehimer[1], Hamdhani Hamdhani[1,2] and Kelsey Hollien[1]

[1] School of Natural Resources and the Environment, University of Arizona, Tucson, AZ, USA
[2] Department of Aquatic Resources Management, University of Mulawarman, Samarinda, Indonesia

## ABSTRACT

**Background:** Aquatic ecosystems are greatly altered by urban development, including the complete loss of natural habitat due to water diversions or channel burial. However, novel freshwater habitats also are created in cities, such as effluent-dependent streams that rely on treated wastewater for flow. It is unclear how diverse these novel ecosystems are, or how quickly aquatic species are able to colonize them. In this study, we (1) quantify odonate (Insecta, Odonata) colonization of a novel effluent-dependent river reach, (2) examine how drying events affect odonates in these novel habitats, and (3) explore whether effluent-dependent streams can support diverse odonate assemblages.

**Methods:** We conducted monthly odonate surveys at three sites along the Santa Cruz River (Tucson, AZ, USA) between June 2019 and May 2020. One site was in a long-established effluent-dependent reach (flowing since the 1970s) that served as a reference site and two sites were in a newly-established reach that began flowing on June 24, 2019 (it was previously dry). We compared odonate species richness, assemblage composition, and colonization patterns across these reaches, and examined how these factors responded to flow cessation events in the new reach.

**Results:** Seven odonate species were observed at the study sites in the new reach within hours of flow initiation, and species rapidly continued to arrive thereafter. Within 3 months, species richness and assemblage composition of adult odonates were indistinguishable in the new and reference reaches. However, drying events resulted in short-term and chronic reductions in species richness at one of the sites. Across all three sites, we found over 50 odonate species, which represent nearly 40% of species known from the state of Arizona.

**Discussion:** Odonates were surprisingly diverse in the effluent-dependent Santa Cruz River and rapidly colonized a newly established reach. Richness levels remained high at study sites that did not experience drying events. These results suggest that consistent discharge of high-quality effluent into dry streambeds can be an important tool for promoting urban biodiversity. However, it remains to be seen how quickly and effectively less vagile taxa (e.g., mayflies, caddisflies) can colonize novel reaches. Effluent-dependent urban streams will always be highly managed systems, but collaboration between ecologists and urban planners could help to maximize aquatic biodiversity while still achieving goals of public safety and urban development.

Corresponding author
Michael T. Bogan,
mbogan@email.arizona.edu

# INTRODUCTION

Urbanization is generally associated with dramatic alterations of natural ecosystems and subsequent losses of biodiversity (*Grimm et al., 2008*). Urban ecosystems frequently have reduced species richness and are more homogeneous when compared to natural systems (e.g., *McKinney, 2006*; *Ball-Damerow, M'Gonigle & Resh, 2014*; *Villalobos-Jimenez, Dunn & Hassall, 2016*). Freshwater ecosystems are particularly affected by urbanization. For example, the physical structure of water bodies is often modified to allow development close to water while trying to reduce flood risk (*Stein et al., 2013*; *Steele & Heffernan, 2014*). Additionally, runoff from impervious surfaces alters flow regimes and hydroperiods, and often delivers a complex combination of excess nutrients and contaminants (*Walsh et al., 2005*). Furthermore, many urban water bodies simply disappear when they are hidden under concrete (*Napieralski & Carvalhaes, 2016*) or completely dry up due to water withdrawals (*Webb et al., 2014*).

However, urbanization also can lead to the creation of novel anthropogenic water bodies, such as canals, stormwater runoff basins, and ponds in urban parks. In some cases, these novel habitats may become important reservoirs of biodiversity, supporting unique communities or species of conservation concern (*Chester & Robson, 2013*; *Lambert & Donihue, 2020*). Aquatic species must be able to colonize these novel habitats via flow connections with adjacent water bodies or overland dispersal. But how quickly do species find these habitats and successfully colonize them? And are the communities that develop therein unique, or are they similar to those from natural habitats or older urban habitats? Novel urban waters could provide an intriguing window into community assembly processes, but these systems were mostly ignored by ecologists for much of the 20th century (*Grimm et al., 2008*). One challenge is that rigorous study of these systems requires coordination between urban planners and ecologists, which is not common practice (*Hunter & Hunter, 2008*; *Lambert & Donihue, 2020*).

Effluent-dependent streams, which rely on discharge from wastewater treatment plants for their baseflow, are increasingly common in urban areas (*Hamdhani, Eppehimer & Bogan, 2020*). In fact, effluent discharge in arid regions has restored flow to some urban streams that were dry for decades because of groundwater overdraft and upstream diversions (*Webb et al., 2014*). Despite decades of research regarding water quality in these effluent-dependent streams (*Brooks, Riley & Taylor, 2006*), only recently have we begun to study their potential biodiversity value (*Bischel et al., 2013*; *Luthy et al., 2015*; *Peschke et al., 2019*). Wastewater treatment plants often discharge effluent into riverbeds simply because there is nowhere else to put it—and it is not customary to notify ecologists when discharge begins or ends (*Brooks, Riley & Taylor, 2006*; *Hamdhani, Eppehimer & Bogan, 2020*).

In this study, we document colonization and community assembly of dragonflies and damselflies (hereafter, odonates) in a new effluent-dependent reach of the Santa Cruz River

in Tucson, AZ (USA). We compare community structure in this new reach with that from a nearby effluent-dependent reach that had been flowing for decades. This rare study opportunity arose because urban planners publicly announced they would begin discharging effluent into the new reach several months prior to flow initiation. We asked three primary research questions: (1) how quickly do dragonflies and damselflies (odonates) colonize novel habitat? (2) can effluent-dependent streams support diverse odonate assemblages? and (3) how do drying events in these novel habitats affect odonates? The third question arose because occasional periods of infrastructure or channel maintenance can lead to cessation of discharge in effluent-dependent streams.

We focused our surveys on odonates for several reasons. First, they tend to be strong dispersers and are known to colonize novel anthropogenic habitats (*Corbet, 1999*; *Prescott & Eason, 2018*; *Cerini, Bologna & Vignoli, 2019*). Second, diverse assemblages of odonates have been found in urban water bodies in many cities (*Willigalla & Fartmann, 2012*; *Goertzen & Suhling, 2015*; *Holtmann et al., 2018*). Third, odonates can be visually surveyed and easily identified to species. Fourth, odonates can be used for biotic indices that assess environmental conditions and ecological integrity in streams (*Chovanec et al., 2015*; *Golfieri et al., 2016*; *Vorster et al., 2020*). Finally, they are conspicuous, colorful, and charismatic, which are useful traits for environmental education and ecotourism (*Lemelin, 2007*; *Clausnitzer et al., 2017*). These potentially important cultural and economic links to urban residents could inspire more collaborations between planners and ecologists to study novel water bodies.

## MATERIALS AND METHODS

### Study system

Historically, the Santa Cruz River had alternating sections of perennial, intermittent, and ephemeral flow throughout its course, from its headwaters in southern Arizona and northern Sonora (Mexico) to its confluence with the Gila River near Phoenix, Arizona (*Webb et al., 2014*). However, diversions and groundwater pumping caused 99% of the river to become ephemeral by the 1940s; groundwater levels have fallen as far as 80 m below the riverbed in Tucson (*Carlson et al., 2011*). Discharge of effluent (treated municipal wastewater) into the dry riverbed has occurred in two reaches of the lower Santa Cruz River since at least the 1970s, restoring perennial surface flow in those sections (Fig. 1). Although water quality in these effluent-dependent reaches was initially very poor, treatment plants were upgraded in 2013 and have produced high quality tertiary-treated effluent ever since (*Sonoran Institute, 2017*).

In 2019, a third effluent-dependent reach of the river was created in downtown Tucson as part of the Santa Cruz River Heritage Project (Fig. 1). The purpose of this new reach is to enhance recharge of the local aquifer and create a new recreational, ecological, and economic feature in the city (*Tucson Water, 2020*). For this new reach, effluent is piped from treatment plants in the north to an outfall location 10 km south of the two established reaches (Fig. 1). Flow began in this new reach on 24 June 2019, but occasional flow reduction or cessation events occurred in the following months as operational issues arose or infrastructure upgrades were needed (Fig. S1). In May 2020, effluent releases ceased in

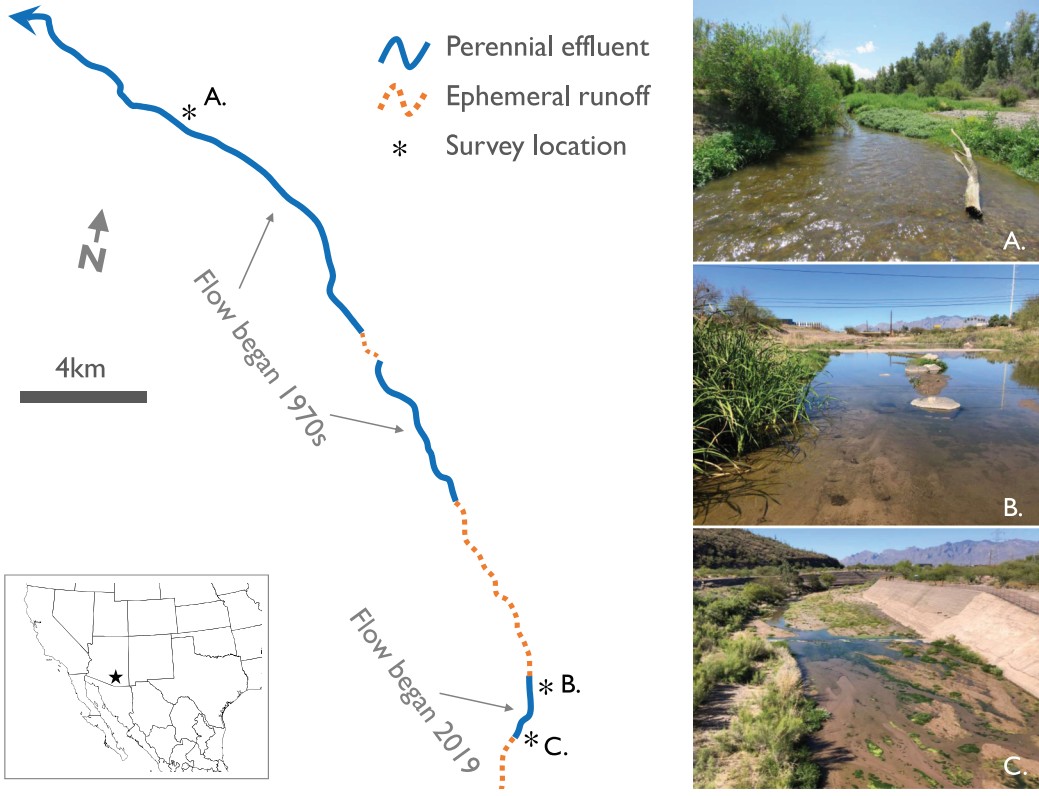

**Figure 1 Map with photos of study sites along established and new effluent-dependent reaches of the Santa Cruz River.** Map of ephemeral and effluent-dependent perennial reaches of the lower Santa Cruz River in and near Tucson, AZ, USA. Two separate reaches in the northern part of the region have been flowing since the 1970s, while the Heritage Project established a new effluent-dependent reach in the southern part of the region in June 2019. Photos illustrate the three study sites: (A) a reference site on a long-established portion of the river, and two sites on the new reach at (B) Cushing Street and (C) Starr Pass Boulevard.

the new reach to allow for sediment removal from the flood control channel, but there are plans to resume flow later in the year (*Tucson Water, 2020*). The nearest naturally perennial stream to all three effluent-dependent reaches is Sabino Canyon, a canyon-bound headwater stream 23 km to the east. The nearest low gradient naturally perennial rivers are the San Pedro and Gila Rivers, >70 km to the east and northeast, respectively.

## Sampling design

We selected three study sites on the Santa Cruz River to survey for adult odonates: (1) a reference site on one of the long-established effluent-dependent reaches, (2) a site near the effluent outfall of the new reach (Starr Pass Blvd), and (3) a site 2 km downstream of the outfall (Cushing St) near the end of the new reach (Fig. 1). The reference site was selected because it supports the highest diversity of aquatic invertebrates (including larval odonates) known from various biomonitoring sites along the established reaches (*Sonoran Institute, 2017*; *Eppehimer et al., 2020*).

At each of the three survey sites, we established 400 m long transects to survey for adult odonates. This length of transect was chosen to incorporate multiple distinct habitat units in one transect to maximize the potential for odonate species richness. In each site, there was at least one riffle, one pool, one run, and one backwater (i.e., off-channel isolated pool), and usually at least two of each, but we did not quantify the exact proportion of each habitat unit within sites. Each of the three sites was located within flood control levees, so while there was a small amount of sinuosity in the actively flowing channel at each site, meandering was limited by levees (Fig. 1).

From 1 June 2019 to 10 May 2020, we conducted monthly 90-min adult odonate surveys along each transect, always between 11:00 and 15:00 h, under partly sunny to clear skies, with minimal winds and warmer temperatures (>15 °C). Although ≥80% of odonate species in a site can be detected with shorter surveys (e.g., 30 min), additional survey time allows greater detection of rare species (Bried et al., 2012). During each survey, the same observer (MTB) slowly walked along and in the water through the 400 m reach while identifying adult odonates by sight and occasionally using a hand-net. Voucher photos for each species observed at each site were taken using a digital camera (Canon Powershot SX60 with 65× optical zoom) and shared with regional odonate experts for taxonomic confirmation (Bailowitz, Danforth & Upson, 2015). Additionally, during each survey, each odonate species observed was scored into one of three abundance categories: (1) rare: <10 individuals seen and no breeding activity observed; (2) uncommon: 10–100 individuals and breeding activity observed; (3) common: >100 individuals with breeding activity observed. There were no instances where less than 10 individuals of a species were seen but breeding was observed among those few individuals. During each survey across the 10 months, larval exuviae and teneral adults were also sought to confirm that a successful reproduction event occurred for a given species observed as adults at a given site.

Surveys at each site generally were conducted on consecutive days within a month, but occasionally occurred as far as 4 days apart so that weather conditions were optimal for each survey. In addition to monthly surveys at all sites, biweekly surveys were conducted at one site (Starr Pass Blvd) on the Heritage reach. These additional surveys were intended to detect new or rare species as soon as possible when they colonized this newly flowing reach, because biweekly surveys can reveal species that monthly surveys fail to capture (Bried, Hunt & Worthen, 2007). During each survey, we recorded weather conditions and measured air temperature and a suite of physiochemical water quality factors (e.g., temperature, conductivity, pH, dissolved oxygen).

Finally, to create a list of all odonate species currently known from the lower Santa Cruz River in the metro Tucson region, and provide context for the results of our surveys, we compiled records from regional experts (R. Bailowitz, D. Danforth & P. Deviche, 2019–2020, personal communications), a regional field guide (Bailowitz, Danforth & Upson, 2015), and online databases (Arizona Dragonflies (azdragonfly.org), iNaturalist (inaturalist.org), and Odonata Central (odonatacentral.org)).

## Data analyses

Survey data were used to calculate species richness values for each site by month. These data were plotted to examine changes in species richness over time at the reference and newly flowing sites, and to see how richness values responded to occasional stream drying events. We also use the Hill numbers approach (*Chao et al., 2014*) to extrapolate the full species richness of each site if additional sampling events were to occur. Estimates of the first Hill number ($q = 0$ for species richness) were made with the package iNEXT (version 2.0.19) in R Version 3.5.3 (*Hsieh, Ma & Chao, 2016*). To ensure fair comparisons, we used only monthly survey data to generate richness estimates, excluding the additional biweekly survey data from the Starr Pass site.

Differences in adult odonate assemblage composition across all sites and survey dates were visualized with non-metric multidimensional scaling (NMS) in PC-ORD Version 5 (MJM Software, Gleneden Beach, OR, USA) using Sorensen distance as the measure of community dissimilarity (*McCune & Grace, 2002*). For ordination analyses, we used survey abundance codes (0 = undetected, 1 = rare, 2 = uncommon, 3 = common) so that species with higher abundances would have more influence on the ensuing ordination than rare species. To assess which species were most influential in the observed ordination patterns, we calculated linear correlation values between species abundances and ordination axes. We also calculated linear correlation coefficients between measured environmental variables and ordination axes. Finally, we examined temporal trends in abundance for each species in the newly flowing reach to visualize colonization patterns in this novel habitat.

## RESULTS

Within 6 h of flow initiation in the newly flowing sites (on 24 June 2019), 7 odonate species (*Enallagma civile*, *Erythrodiplax basifusca*, *Ischnura demorsa*, *Orthemis ferruginea*, *Pachydiplax longipennis*, *Sympetrum corruptum*, and *Tramea onusta*) were observed mating and ovipositing (Fig. 2). Additional species quickly arrived in the following weeks, and species richness at the Starr Pass site reached or exceeded that of the reference site within 3 months (Fig. 3). Species richness values at the Cushing site, which experienced several drying events during the course of the study, remained lower than Starr Pass but exhibited similar seasonal trajectories (i.e., lower in winter months, higher in spring and summer months). Species richness in both of the newly flowing sites plummeted when flow ceased in May 2020 (Fig. 3). The Hill numbers estimation of species richness found almost complete overlap in 95% confidence intervals for the Starr Pass and reference sites, but values at Cushing were predicted to remain lower even with numerous additional monthly sampling events (Fig. 4). Across all sites and dates, we observed 50 of the 53 odonate species currently known from the entire lower Santa Cruz River, including 44 species at the reference site, 43 species at Starr Pass, and 28 species at Cushing (Table 1). We confirmed successful recruitment via larval exuviae and/or teneral adults for 35 species at the reference site (80%), 28 species at Starr Pass (65%), and 17 species at Cushing (61%).

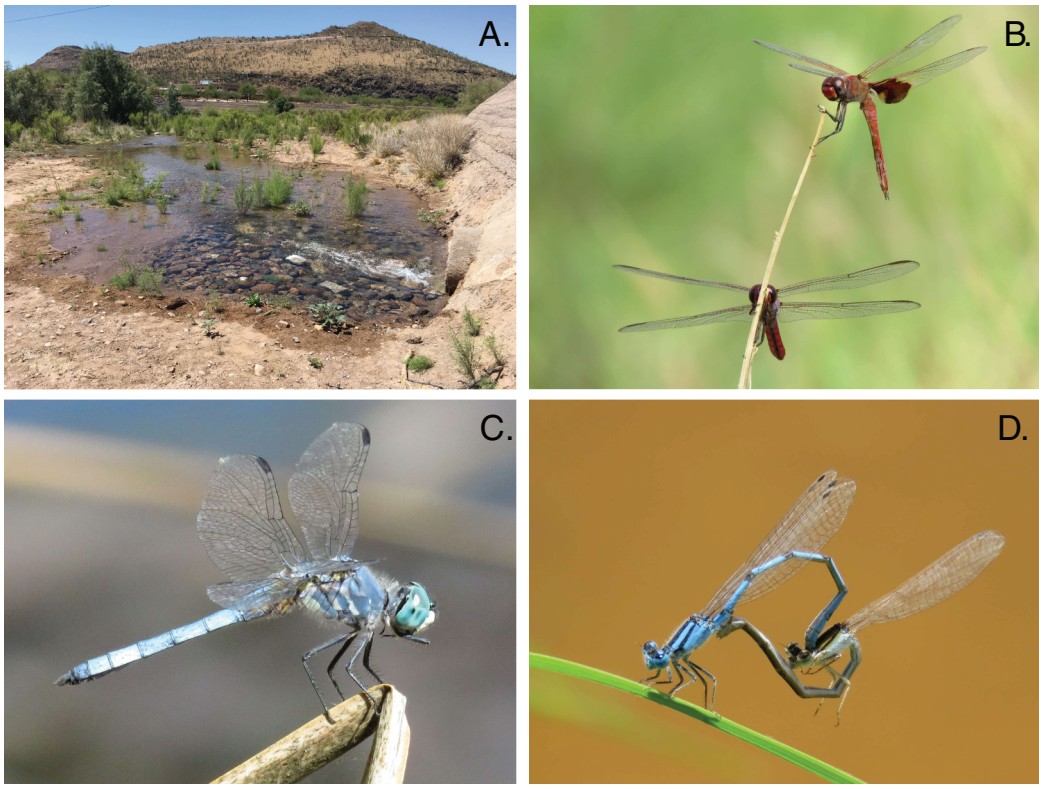

**Figure 2 First day of flow initiation and early odonate colonists in the Santa Cruz River.** Photos from the first day that treated effluent was released into the previously-dry channel of the Santa Cruz River as part of the Heritage Project in Tucson, AZ (A) Odonate colonists arriving within 6 h of flow initiation included Red Saddlebags (*Tramea onusta*) and Roseate Skimmers (*Orthemis ferruginea*) (B), Blue Dashers (*Pachydiplax longipennis*) (C), and Familial Bluets (*Enallagma civile*) (D).

Assemblage composition between Star Pass and the reference site converged within 3 months (Fig. 5; NMS results: stress = 0.14, final instability > 0.00001, cumulative $R^2$ = 0.84, $p$ = 0.004). Surveys from the reference site generally occupied the lowest region of NMS axis 2, remained stable through summer 2019, and were characterized by high abundances of several damselflies (e.g., *Argia sedula*, *Hetaerina americana*, *Argia pallens*, *Telebasis salva*) and dragonflies (e.g., *Anax junius*, *Libellula saturata*) (Table 2). Assemblages at all three sites moved to right along NMS axis 1 for the winter months, exhibiting lower abundances of monsoonal or summer dragonflies (e.g., *Orthemis ferruginea*, *Pantala flavescens*, *Tramea lacerata*) and two species of damselflies (*Ischnura demorsa*, *Enallagma civile*) (Table 2). Overall, the reference site exhibited less variation in assemblage composition through time than the two newly flowing sites (Fig. 5). Both water and air temperatures were strongly negatively correlated with NMS axis 1 ($r$ = −0.77 and −0.73, respectively); all other measured variables were only weakly correlated with ordination axes (i.e., −0.5 < $r$ < 0.5).

Of all species known to have breeding populations in the reference site, only 6 failed to become established at the new sites within 10 months (Table 1). We also found 7 species in the newly flowing sites that were absent from the reference site; however, 6 of these were

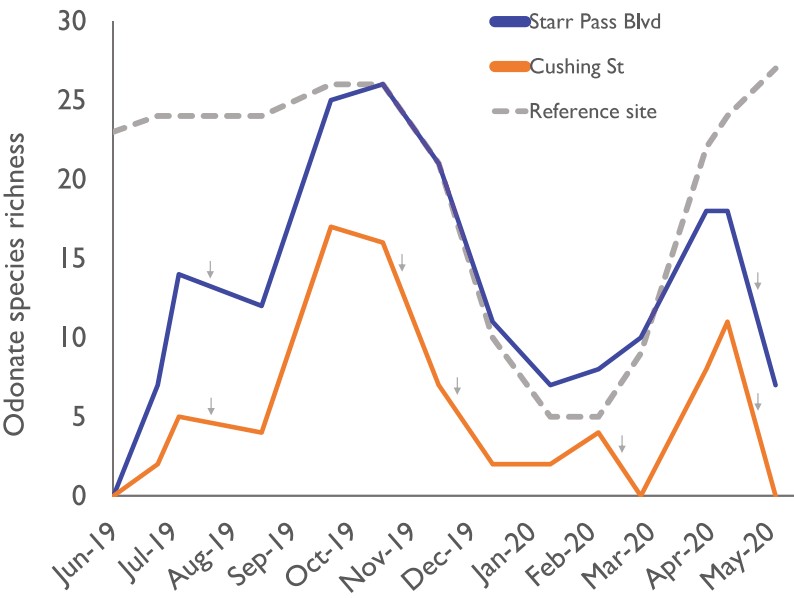

**Figure 3 Odonate species richness across the 10-month study period at three sites along the effluent-dependent Santa Cruz River.** Odonate species richness values observed during monthly surveys at two sites (Starr Pass Blvd and Cushing St) sites in the newly flowing reach of the Santa Cruz River, and at a reference reach downstream that has been flowing with effluent since the 1970s. Small gray arrows indicate times when a site dried up due to reduced effluent discharge into the new reach.

rare and were never observed breeding. Only one species of damselfly (*Argia extranea*) became established at the new sites but was never observed at the reference site. Several different colonization modes were observed among the species documented in the newly flowing sites (Fig. 6). Some species colonized on the first day of flow initiation, quickly established robust breeding populations, and remained abundant through the course of the study, while other species colonized rapidly but exhibited seasonal variation in abundances thereafter. Yet other species took a few months to colonize and remained at relatively low abundances, which varied seasonally. Finally, some species were vagrants that only appeared once or twice and did not appear to have breeding populations in any of the sites (e.g., *Libellula luctosa*).

## DISCUSSION

Dragonflies and damselflies colonized a new effluent-dependent reach of the Santa Cruz River incredibly fast, with seven species arriving on the first day of flow, each of which successfully established breeding populations. This was especially surprising because flow began during the hottest and driest part of the year, and many aquatic insects in the region only disperse aerially after summer rains begin (*Bogan & Boersma, 2012*). Within 3 months, species richness values of adult odonates were equal in the new reach and the long-established reach more than 10 km away. Although previous studies have found that odonates can disperse tens to hundreds of kilometers across arid landscapes (*Suhling, Martens & Suhling, 2017*), the speed at which species arrived here was still impressive. It often takes >2–3 months for odonates to find novel water bodies, with lower

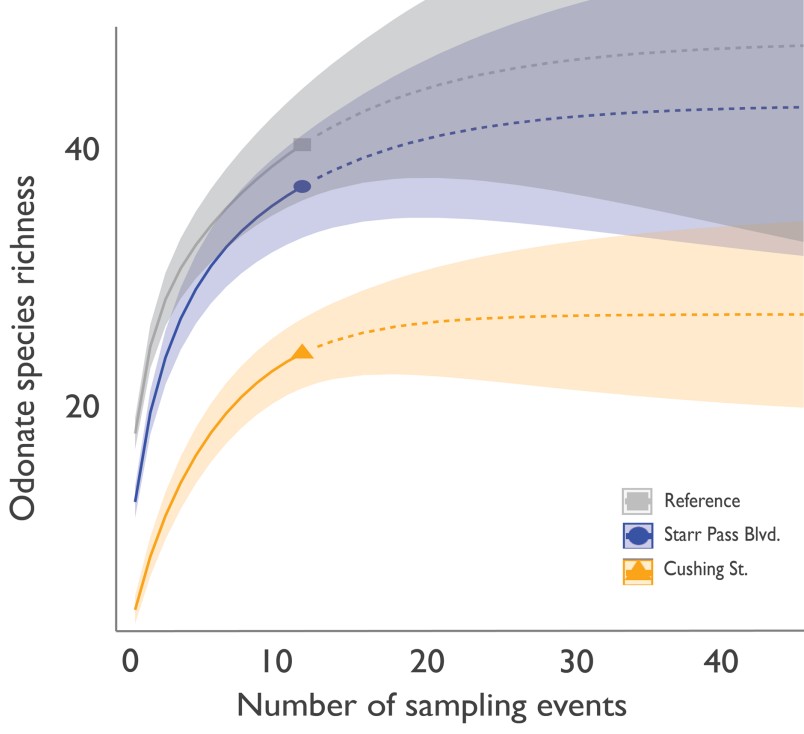

**Figure 4 Interpolated and extrapolated values for cumulative odonate species richness at three sites along the effluent-dependent Santa Cruz River.** Cumulative odonate species richness by site after each monthly sampling event (solid lines: interpolated) and projected values (dashed lines: extrapolated) if additional sampling months were to occur, as predicted using Hill numbers for species richness ($q = 0$). Shaded areas represent a 95% confidence interval.

colonization rates usually observed at more isolated waters (*McCauley, 2006*; *Bogan & Boersma, 2012*; *Groover, 2017*).

Both the rapid colonization rates and the high overall species richness values observed were surprising. Across all dates, we found 50 odonate species in our three effluent-dependent study sites, which is >35% of the species known from the entire state of Arizona (*Bailowitz, Danforth & Upson, 2015*). For comparison, only 58 species were found in the entire Grand Canyon (*Stevens & Bailowitz, 2009*), an area is many times larger than our study area and encompasses many desert and montane springs, streams, and rivers. Odonates thrive in warm environments (*Corbet, 1999*), so the warm air temperatures and mild winters of southern Arizona may be partly responsible for the high odonate diversity we observed. Although nearly 100 odonate species are known from Pima County, where the lower Santa Cruz River is located, the county hosts aquatic habitats in areas ranging from hot deserts (<275 m asl) to cold montane forests (>2,700 m asl) (*Bailowitz, Danforth & Upson, 2015*). So, the fact that over half of the species known in the county were found in a single, effluent-dependent urban river—an artificial ecosystem—is remarkable.

We could not compare our findings to those from naturally perennial rural reaches of the river outside of Tucson because none remain (*Webb et al., 2014*). However, nearly 70% of published studies report some reduction in odonate species richness in urban cores,

**Table 1 Odonate species encountered at three sites along the effluent-dependent Santa Cruz River.**
Each species was scored into one of three abundance categories for each site: (1) rare: <10 individuals seen; (2) uncommon: 10–100 individuals; (3) common: >100 individuals. We observe evidence of reproduction success at a given site (e.g., teneral adults) for species classified as common or uncommon, but not for those classified as rare. Species marked with an asterisk were not found during the surveys for this study, but have been documented from adjacent reaches of the Santa Cruz River in existing databases (e.g., iNaturalist, Odonata Central).

| Suborder | Family | Species | Reference | Starr Pass Blvd | Cushing St |
|---|---|---|---|---|---|
| Anisoptera | Aeshnidae | *Anax junius* | Common | Common | Common |
| | | *Anax walsinghami* | Rare | Rare | |
| | | *Rhionaeschna multicolor* | Common | Rare | Rare |
| | Gomphidae | *Erpetogomphus compositus* | Common | Rare | |
| | | *Erpetogomphus lampropeltis* | | Rare | |
| | | *Progomphus borealis* | Common | Common | Rare |
| | | *Stylurus plagiatus** | | | |
| | Libellulidae | *Brachymesia furcada** | | | |
| | | *Brechmorhoga mendax* | Uncommon | Uncommon | |
| | | *Dythemis nigrescens* | Uncommon | Rare | |
| | | *Erythemis collocata* | Common | Common | Uncommon |
| | | *Erythemis vesiculosa* | Rare | | |
| | | *Erythrodiplax basifusca* | Common | Common | Uncommon |
| | | *Libellula forensis* | Rare | | |
| | | *Libellula luctuosa* | Rare | | Rare |
| | | *Libellula pulchella* | Rare | Rare | Rare |
| | | *Libellula saturata* | Common | Common | Uncommon |
| | | *Macrothemis inacuta* | Uncommon | Uncommon | |
| | | *Orthemis ferruginea* | Common | Common | Common |
| | | *Pachydiplax longipennis* | Common | Common | Rare |
| | | *Paltothemis lineatipes* | | Rare | |
| | | *Pantala flavescens* | Common | Common | Uncommon |
| | | *Pantala hymenaea* | Uncommon | Uncommon | Uncommon |
| | | *Perithemis intensa* | Common | Common | Rare |
| | | *Plathemis lydia** | | | |
| | | *Pseudoleon superbus* | Uncommon | Uncommon | Rare |
| | | *Sympetrum corruptum* | Common | Common | Common |
| | | *Sympetrum illotum* | | Rare | |
| | | *Tramea lacerata* | Common | Common | Uncommon |
| | | *Tramea onusta* | Uncommon | Common | Uncommon |
| Zygoptera | Calopterygidae | *Hetaerina americana* | Common | Uncommon | |
| | Coenagrionidae | *Argia extranea* | | Common | |
| | | *Argia immunda* | Uncommon | Rare | |
| | | *Argia moesta* | Common | Rare | Rare |
| | | *Argia nahuana* | Uncommon | Common | Rare |
| | | *Argia pallens* | Common | Common | Uncommon |
| | | *Argia sedula* | Common | Common | Uncommon |

| Suborder | Family | Species | Reference | Starr Pass Blvd | Cushing St |
|---|---|---|---|---|---|
| | | *Argia tezpi* | Rare | Rare | |
| | | *Argia translata* | Uncommon | | |
| | | *Enallagma basidens* | Common | | |
| | | *Enallagma civile* | Common | Common | Common |
| | | *Enallagma eiseni* | Rare | | |
| | | *Enallagma novaehispanae* | Common | Rare | |
| | | *Enallagma praevarum* | Common | Common | Rare |
| | | *Enallagma semicirculare* | Rare | | |
| | | *Hesperagrion heterodoxum* | | Rare | |
| | | *Ischnura cervula* | Uncommon | Uncommon | |
| | | *Ischnura demorsa* | Common | Common | Common |
| | | *Ischnura hastata* | Common | Common | Uncommon |
| | | *Ischnura ramburii* | Common | Common | Uncommon |
| | | *Telebasis salva* | Common | Common | Common |
| | Lestidae | *Archilestes californicus* | Rare | Rare | |
| | | *Lestes alacer* | | Rare | Rare |

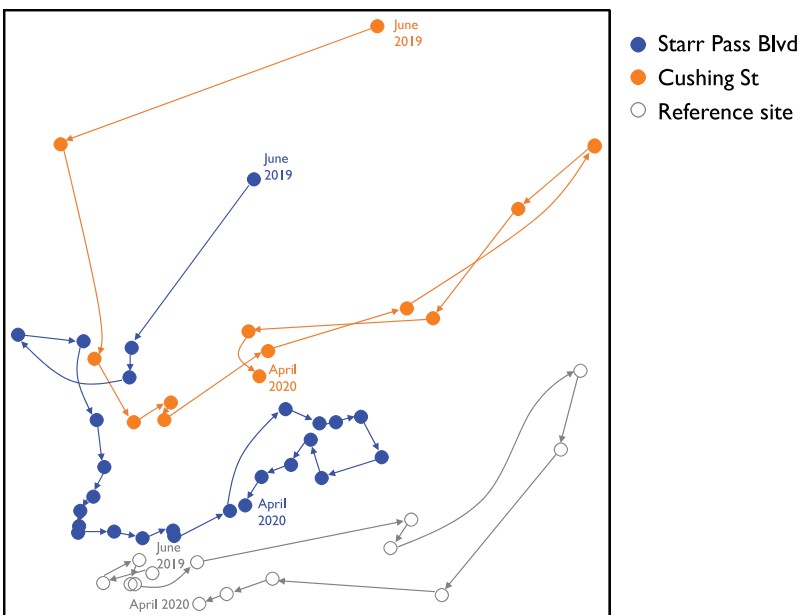

**Figure 5 Ordination plot of changes in odonate assemblages over time at three study sites along the effluent-dependent Santa Cruz River.** NMS ordination plot visualizing odonate assemblage composition from all surveys at the three study sites along the Santa Cruz River from June 2019 to April 2020. Vectors link subsequent surveys at each site and illustrate trajectories of compositional changes through time.

with species losses often being related to pollution issues (*Villalobos-Jimenez, Dunn & Hassall, 2016*). Interestingly, many of these studies reported reductions in odonate richness in ponds rather than rivers. For example, a study from Kentucky (USA) found no

**Table 2 Correlations between odonate species abundances and NMS ordination axis values.** Pearson's correlation coefficients between abundances of individual odonate species and NMS axis 1 and 2. Species only showed strongly negative correlation values ($<-0.5$) with either or both axes; no strongly positive correlation values ($>0.5$) were observed.

| Species | Axis 1 $r$ | Species | Axis 2 $r$ |
|---|---|---|---|
| *Orthemis ferruginea* | −0.79 | *Argia sedula* | −0.78 |
| *Pantala flavescens* | −0.69 | *Hetaerina americana* | −0.67 |
| *Ischnura demorsa* | −0.68 | *Argia pallens* | −0.67 |
| *Tramea lacerata* | −0.61 | *Anax junius* | −0.66 |
| *Libellula saturata* | −0.60 | *Telebasis salva* | −0.64 |
| *Erythrodiplax basifusca* | −0.57 | *Libellula saturata* | −0.61 |
| *Erythemis collocata* | −0.56 | *Enallagma novaehispanae* | −0.61 |
| *Tramea onusta* | −0.51 | *Sympetrum corruptum* | −0.60 |
| *Enallagma civile* | −0.51 | *Perithemis intensa* | −0.58 |
| *Anax junius* | −0.50 | *Ischnura ramburii* | −0.58 |
| | | *Argia moesta* | −0.53 |

difference in odonate diversity between urban and rural streams, but urban ponds were less diverse than rural ponds (*Prescott & Eason, 2018*). The high diversity we observed in the Santa Cruz River also could be due in part to the relative lack of industrial pollution (e.g., factories) and high-density development in Tucson. Furthermore, the warm water temperature regimes of effluent-dependent streams (*Bischel et al., 2013*) may be ideal for the growth and development of odonates. In fact, some odonates we observed, such as the Neotropical bluet (*Enallagma novaehispaniae*), are tropical species that have only colonized Arizona in the past decade (*Bailowitz, Danforth & Upson, 2015*). To date, they have only been found in effluent-dependent streams, whose warm waters may mimic their tropical home streams (R. Bailowitz & P. Deviche, 2020, personal communication). Similar patterns have been observed in central European cities, where southern Mediterranean odonate species have been documented expanding their range northward (*Willigalla & Fartmann, 2012*).

Although we found diverse odonate assemblages in the tertiary-treated wastewater of the Santa Cruz River, this does not mean that wastewater is always beneficial to odonates. Untreated wastewater (i.e., raw sewage) generally reduces the abundance and diversity of odonates (*Henriques-de-Oliveira, Baptista & Nessimian, 2007*). In fact, within wastewater treatment facilities, odonate species richness and abundance increase dramatically from initial wastewater lagoons to subsequent treatment ponds with better water quality (*Catling, 2005*). Given these sensitivities to organic pollution, odonates are often used as bioindicators (*Júnior, Juen & Hamada, 2015*; *Mendes et al., 2017*; *Miguel et al., 2017*). Some species are known to tolerate inputs of raw or poorly treated sewage (e.g., *Anax junius*, *Enallagma civile*: *Catling, 2005*). Unfortunately, there are no historical odonate data available from our reference site on the Santa Cruz River when it was receiving lower quality effluent (1970–2013). However, studies from the upper Santa Cruz River, 60 km to the south of our study sites, may be informative. *Boyle & Fraleigh (2003)*

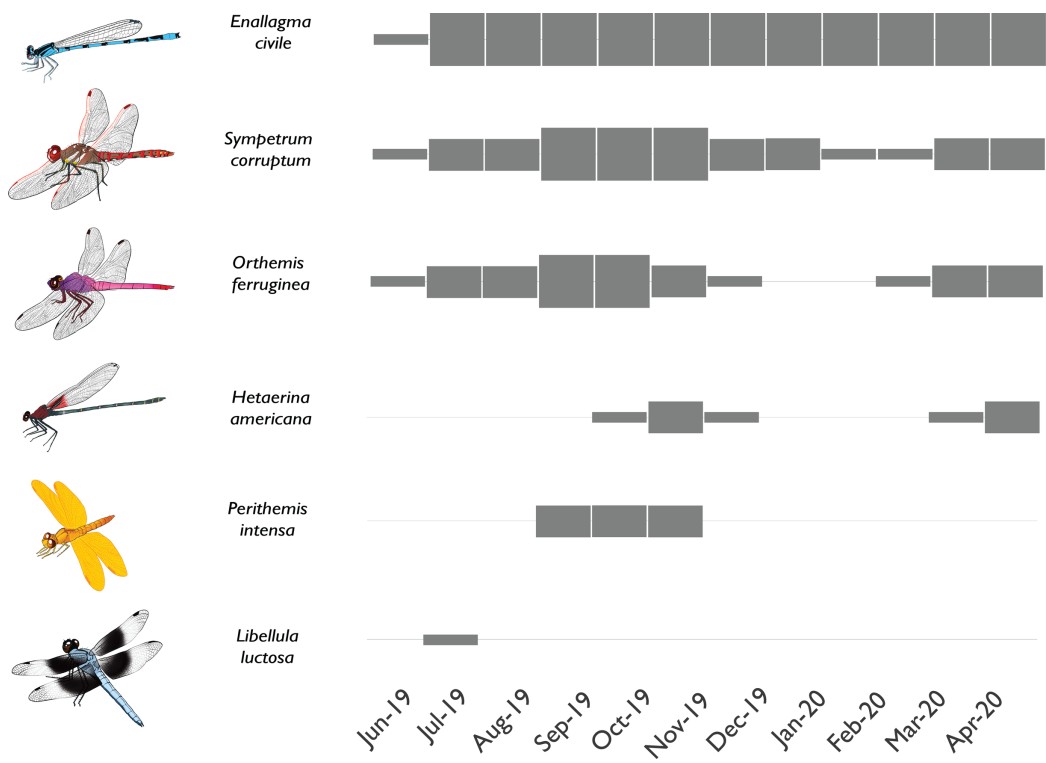

**Figure 6 Phenology of selected odonate colonists in a newly-flowing reach of the effluent-dependent Santa Cruz River.** Phenology plot illustrating the presence and abundance of different odonate species through time in the newly flowing Heritage reach of the Santa Cruz River (thin gray box = rare; medium gray box = uncommon but breeding; thick gray box = common and breeding). Some species rapidly colonized the new reach quickly and were abundant through the study (e.g., *Enallagma civile*) or were at least abundant during warmer weather (e.g., *Sympetrum corruptum*, *Orthemis ferruginea*). Other species took longer to colonize and had limited seasons of adult flight activity (e.g., *Hetaerina americana*, *Perithemis intensa*). Other species were vagrants that appeared once or twice but failed to establish populations (e.g., *Libellula luctosa*).

found only six odonate genera in the upper Santa Cruz River when it was fed by low quality effluent in the 1990s. In contrast, we found 27 genera in the lower Santa Cruz. These findings suggest that wastewater treatment plant upgrades are at least partly responsible for the high odonate diversity we observed.

One risk of living in effluent-dependent streams is that drying events can occur when infrastructure fails (e.g., pipes break) or discharge is paused to allow for channel maintenance (*Tucson Water, 2020*). In naturally flowing streams, increased frequency or duration of drying events usually causes reductions in the diversity of aquatic invertebrates (*Datry et al., 2014*; *Stubbington et al., 2017*). For odonates, drying events eliminate larvae of most species, and these losses will have cascading impacts on adult populations (*McPeek, 2008*). In our study, we observed much higher richness of adult odonates in the site that dried infrequently versus the one that dried multiple times (Fig. 3). However, short periods of flow cessation (e.g., hours to a couple days) did not seem to have a dramatic effect, as odonate larvae likely found refuge in damp algal mats or remnant pools (*Stubbington et al., 2017*). To maximize odonate diversity in effluent-dependent

streams, managers should minimize the duration and frequency of shutoff events that result in stream drying, and avoid shutting off flow during the hottest, driest times of the year, when in-stream refuges would quickly disappear.

Although recurring drying events are likely the primary cause of reduced species richness observed at the Cushing site, that site's species richness trajectory was lower than that of the nearby Starr Pass even before drying began (Fig. 3). One potentially important factor that we did not measure is the structural complexity of riparian vegetation. Neither site had well-developed mesophilic riparian vegetation; however, the Starr Pass site included the effluent outfall, where discharge flowed for ~60 m across a vegetated terrace (Fig. 2) before dropping into the active river channel. This area may have provided an enhanced number of perches or structural complexity that were "attractive" to odonates (e.g., *Samways & Steytler, 1996*). After several months of flow, however, the complexity of riparian vegetation seemed to increase at both sites as wetland plants colonized the river (e.g., *Typha* sp.: Fig. 1). Future studies of new effluent-dependent stream reaches should quantify riparian vegetation complexity before and after flow begins.

One major limitation of our study is that there are no historical data available from when our study reaches were naturally perennial in the early 1900s. We do not know what the diversity or composition of odonates was in the natural Santa Cruz River. However, studies from other regions suggest that the lower Santa Cruz River, as diverse as it is today, may still lack species which were present historically. For example, the odonate fauna of California today is more homogeneous than it was 100 years ago, with sharp losses of habitat specialist species in urbanized areas (*Ball-Damerow, M'Gonigle & Resh, 2014*). Additionally, we know that vegetation along the Santa Cruz River has changed dramatically in the last century, with the elimination of floodplain forests and the extirpation of many native plant species (*Webb et al., 2014*). Odonate diversity generally increases with the diversity and complexity of riparian vegetation in both urban and rural habitats (*Samways & Steytler, 1996*; *Goertzen & Suhling, 2013*; *Dutra & De Marco, 2015*). Thus, historic vegetation losses along the Santa Cruz River may have led to the extirpation of some odonate species that were historically present.

Finally, it is likely that odonate recolonization of the Santa Cruz River is still happening. The river was dry for many decades in all reaches, and even in the long-established effluent-dependent reaches, water quality has only been high since 2013. So, the river has only been "palatable" to many odonate colonizers for a few years. Further, the nearest naturally perennial stream is a small headwater stream over 20 km away, and the nearest perennial rivers of similar size to the Santa Cruz are over 70 km away. Rare and stochastic colonization events from these distant source streams and rivers may take time and likely are still occurring. For example, the dragonfly *Stylurus plagiatus* is known from natural rivers in Arizona (*Bailowitz, Danforth & Upson, 2015*), but it was not known from the effluent-dependent Santa Cruz River until we found larvae and adults in two reaches in 2019 (M.T. Bogan, 2019, unpublished data). Even over the relatively short duration of the current study, we observed that colonization rate and success varied greatly among odonate species (Fig. 6). Repeat surveys and long-term studies along the

Santa Cruz River will be invaluable for documenting colonization processes in this novel urban ecosystem.

## CONCLUSIONS

Odonates were surprisingly diverse in the effluent-dependent Santa Cruz River, supporting nearly 40% of all species known in the state of Arizona. Additionally, numerous odonate species rapidly colonized a newly-established reach of the river. In the absence of prolonged drying events, assemblage composition in the new sites was indistinguishable from the reference site within 3 months. These results suggest that consistent discharge of high-quality effluent into dry streambeds can be an important tool for promoting urban biodiversity, especially in arid and semi-arid regions (*Bischel et al., 2013*; *Luthy et al., 2015*). However, it remains to be seen how quickly and effectively less vagile taxa (e.g., mayflies, caddisflies) will colonize novel reaches, and this topic deserves further study. Furthermore, our study ended because flow ceased in the newly-established reach to allow for sediment removal and flood risk mitigation (*Tucson Water, 2020*). This dramatic ending highlights the fact that urban streams will always be highly managed systems. But with collaboration between ecologists and urban planners, these management activities can be modified to maximize aquatic biodiversity while still achieving public safety goals (*Hunter & Hunter, 2008*). Collaborations between ecologists and planners also would enhance ecotourism opportunities and better connect urban residents with their local ecosystems (*Lemelin, 2007*; *Clausnitzer et al., 2017*). The 20th century was a difficult time for urban streams and the species that resided in them, but there is hope for better ecological outcomes by the end of the 21st century.

## ACKNOWLEDGEMENTS

The study reaches of the Santa Cruz River are part of the traditional homeland of the Tohono O'odham, and the river runs through portions of both the Tohono O'odham and Pascua Yaqui Nations. Many thanks to Rich Bailowitz, Doug Danforth, and Pierre Deviche for being generous with their time, their data, and their taxonomic expertise. We greatly appreciate the time and cooperation of Dick Thompson and Maya Teychea, hydrologists at Tucson Water who generously shared their time and data. Thanks also to Claire Zugmeyer (Sonoran Institute), Pima County Natural Resources, Pima County Regional Flood Control District, Dan Marries and Paul Durrant (KOLD News), Kate Boersma, and Devlin Houser for their logistical support during the course of this research. Special thanks to Maya Stahl for her odonate art.

### Funding

The authors received no funding for this work.

### Competing Interests

The authors declare that they have no competing interests.

## Author Contributions

- Michael T. Bogan conceived and designed the experiments, performed the experiments, analyzed the data, prepared figures and/or tables, authored or reviewed drafts of the paper, and approved the final draft.
- Drew Eppehimer conceived and designed the experiments, authored or reviewed drafts of the paper, and approved the final draft.
- Hamdhani Hamdhani conceived and designed the experiments, authored or reviewed drafts of the paper, and approved the final draft.
- Kelsey Hollien conceived and designed the experiments, analyzed the data, prepared figures and/or tables, authored or reviewed drafts of the paper, and approved the final draft.

## Field Study Permissions

The following information was supplied relating to field study approvals (i.e., approving body and any reference numbers):

Field surveys were done in coordination with the Pima County Regional Flood Control District, which manages the land the Santa Cruz River flows through. No permits are necessary to study non-endangered invertebrates (including odonates) in the state of Arizona.

## Data Availability

Raw data are available in the Supplemental Files.

## Supplemental Information

Supplemental information for this article can be found online at http://dx.doi.org/10.7717/peerj.9856#supplemental-information.

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
