# Peer review of "If you build it, they will come: rapid colonization by dragonflies in a new effluent-dependent river reach"

_PeerJ, doi:10.7717/peerj.9856_

## Round 0.1 · original submission · Minor Revisions

All three reviewers have indicated minor revisions to your paper and I concur - one specific area requiring attention is the rate of recolonization, and whether the presence of adults at a site is really sufficient evidence for recolonization without also establishing the presence of larvae / exuviae.

Reviewer 1 ·

Basic reporting

The manuscript describes the adult odonate fauna of a stretch of Santa Cruz River in southern Arizona. The authors took advantage of the fact that they could compare odonate diversity (and changes thereof over time) in a section of river with established perennial water flow with that in two sections of the same river where this flow, likewise consisting of treated wastewater, was recently restored and underwent large fluctuations during the study period. Studies such as this have the potential to improve our understanding of how insect communities adjust to urbanization, colonize new habitats, and respond to short- and longer-term human-related environmental manipulations, and they are, therefore, welcome and warranted.

Experimental design

I found the manuscript to be well-written and organized, and informative. The data were analyzed using appropriate statistical methods. The results are novel and contribute to our knowledge of odonate population dynamics and ecology, as well as to our understanding of urban aquatic ecosystems.

Validity of the findings

See above and below.

Additional comments

L252: The rate of colonization (i.e., species showing up within a few hours of water flow restoration) was, indeed, rather fast, but why the authors conclude that species richness (50 species) at the study sites during the survey period was higher than expected is unclear. Approximately 100 odonate species are recorded in Pima Co., where the work was conducted, and so 50% of the species present in the county were cumulatively found during surveys. Survey sites encompass diverse types of micro-habitats and one stretch of river that was investigated has had water consistently running for several years. Given overall regional richness (approx. 100 species) and the diversity of micro-habitats that were surveyed, one would predict a relatively large number of species to show up at least occasionally and this was, indeed, the case. Thus, and in view of the demonstrated dispersal ability of odonates in general, what is the rationale for stating that species richness was higher than anticipated? The authors compare their findings with those in much larger regions of the state (Stevens & Bailowitz 2008, 2009) and where similar numbers of species (32 and 58, respectively) as recorded here were found. That a much smaller survey area had comparable richness is not particularly surprising because the Santa Cruz river is located well south of and at lower altitude than the regions surveyed by Stevens and Bailowitz, and so is expectedly to hold considerably more species to begin with.

L275: The Neotropical Bluet was first found in Arizona in 2010 and so has been present in the state, at least sporadically, for more than 5 years – although it is unknown when breeding populations became established. Too, this bluet is common/abundant in the southern half of Texas and so presumably colonized the United States (L275) quite a while ago.

L301: The authors write “We observed much higher richness of adult odonates in the site that dried infrequently versus the one that dried multiple times”. This is correct based on data shown in Fig. 3. However, this figure also shows that this difference was already present in July 2019, i.e., before the first drying event. Thus, drying does by itself not seem to fully explain the observed site differences in odonate richness, indicating that other factors are likely involved.

I am wondering whether the authors could comment on the possibility that the effects of drying on odonate populations may be season dependent. For example, a short period of drying in early summer, before the summer monsoon and when temperatures are usually elevated, would be expected to have a more pronounced and faster effect than a same period of drying during winter, when ambient temperatures are much lower. If so, water resource managers may want to consider limiting drying to the winter, if possible, in order to minimize negative impacts on aquatic biodiversity.
Minor issues:

L101 & L259: Corbet, 1999

L224: were strongly negatively correlated…

L238: Libellula

L248: Suhling et al., 2017 is not in the list of references.

Fig. 2: Why include two pictures of Enallagma civile pairs (one in wheel, the other in tandem with ovipositing female)? Readers would probably prefer to see pictures of different species here.

Fig. 5. Label X and Y axes.

Supplemental data table(s): Red-tailed Pennant = Brachymesia *furcaTa*; Neotropical Bluet = Enallagma *novaehispanIae*; Western Pondhawk = Erythemis *collOcata*; *FilIgree* Skimmer.

·

Basic reporting

line 189 – I suggest “undetected” in place of “absent.”

lines 252–253 – The authors say that “Not only were colonization rates faster than expected, but overall species richness values were higher than we expected as well.” I do not mean to pedantic, but when I see “expect” in a scientific study I construe it to mean a null expectation (i.e., from a null model) or some statistical expectation. In this case, though, I think the authors mean that the results surprised them, which is not the same thing.

lines 255–258 – I do not find the comparisons to Ash Meadows or the Grand Canyon compelling. They are in different biogeographic provinces and ecoregions, ones that have lower richness across the broad, not just with regard to Odonata.

lines 267–269 – This sentence needs to be reworded. As it stands, it says, “diversity did not differ even though diversity was lower.”

line 270 – I am unsure what is meant by “more robust tolerances of lotic versus lentic odonates.” It is axiomatic that more lotic than lentic species would occur along a river, especially a shallow one with steady flow.

lines 274–275 – What the authors mean here is that Enallagma novaehispaniae has colonized southeastern Arizona in the past five years (or so). It has been known from and well established in the United States for many decades.

Experimental design

If anything gave me pause in this study it revolves around two research questions that intrigued me: “[H]ow quickly do dragonflies and damselflies (odonates) colonize novel habitat?” and “[C]an effluent-dependent streams support diverse odonate assemblages?” Colonization is tricky to establish in Odonata, precisely because odonates are so vagile, the upshot being that just because one can find a suite of odonate species at a site it does not follow that those species have colonized the site. The survey methods perhaps suggest a way around this conundrum, but I cannot be certain. The authors state that they “selected three study sites on the Santa Cruz River to survey for *adult* odonates” [emphasis added] yet later in the subsequent paragraph they imply that they surveyed for “larval emergence” [=tenerals?]. If the latter is true, then I recommend that it is clearer earlier that surveys were not confined to adults. If, however, principally adult surveys alone were conducted, then I suggest the authors adopt some sort of indicator model (e.g., Bried et al. 2015, Freshwater Science; Patten et al. 2019, Ecological Indicators) to support the idea that a breeding population was established.

Validity of the findings

Only in the sense that findings hinge on species having in fact colonized.

Additional comments

no comment

Reviewer 3 ·

Basic reporting

This is an interesting and well-researched paper. However, I have some issues mainly with sampling design and the description of the study system.

Detailed comments

Line 100 to 108 (Introduction)
I would recommend to add another significant reason for focusing on odonate surveys, i.e. that the collected data of odonate assemblages can be summarized in biotic indices. There are valid examples of dragonfly-based biotic indices developed mainly, but not exclusively, for river assessment in different geographical contexts and continents: e.g. central Europe (Chovanec et al., 2015), Mediterranean Europe (Berquier et al., 2016; Golfieri et al., 2016) and Africa (Vorster et al., 2020).

Line 515 (References)
The alphabetical order of this reference is not correct.

Experimental design

Detailed comments

Line 123 to 133 (Study system)
The authors need to better characterize the channel morphology of the study reaches, possibly according to the reach-scale classifications of Rinaldi et al. 2016 (Aquatic sciences) (e.g. sinuous, meandering, wandering, braided etc.) and Brierley and Fryirs, 2005. Moreover, they should also describe channel substrate material and the presence/percentages of aquatic and riparian vegetation in the three sampling sites. Lastly, they could include data about hydromorphological assessment of the study reaches, if available, as morphological conditions significantly influence odonate assemblages at the reach scale (Golfieri et al., 2018).

Line 132 to 133 (Study system)
The authors state that “The nearest naturally perennial stream to all three effluent dependent reaches is Sabino Canyon, 23 km to the east.”, but later on, at lines 324-325 in the Discussion section, they state that “the nearest perennial river source populations are over 100 km away (e.g. the San Pedro and Gila Rivers)”. These statements seem to be contrasting and the authors should clarify this element.

Line 136 (Sampling design)
Samplings focused on adult odonates, but why the collection of larvae and/or exuviae was not considered, at least during the second half of the research period? Are larval stages of all the resident species described and identifiable? Several studies underline the importance of sampling larvae and/or exuviae to characterize odonate assemblages (e.g. Raebel et al., 2010). Moreover, later on, at lines 157 to 159, the authors introduce the issue of “larval emergence” at the site without giving any explanation about how these data were collected and treated. This element has to be clarified.

Line 144 (Sampling design)
Each study transect incorporate multiple distinct habitat units: is it possible to summarize the habitats units sampled in each transect with a table? Could this element (i.e. a difference in the habitat units sampled) influence the different results of the study sites?

Line 155 to 159 (Sampling design)
The authors used three abundance categories, including also elements of breeding behaviour. I would suggest dividing abundance categories from behaviour description, as it also possible to observe less than 10 individuals showing breeding activity. In addition, a clear explanation of the criteria used to consider a species as breeding is needed.

Validity of the findings

Detailed comments

Line 266 to 270 (Discussion)
The authors partly explain the high diversity of Santa Cruz River as a consequence of the “more robust tolerances of lotic odonates versus lentic odonates”. This can be true in some specific contexts, but studies carried out in other countries revealed a significant impact of hydromorphological alterations and related urbanization on riverine odonate assemblages (e.g. Golfieri et al., 2016 and 2018).

Line 288 (Discussion)
The word “junius” has to be written in italics.

---

## Round 0.2 · accepted · Accept

Thanks for your patience with the review process, and for responding directly to all reviewers' criticisms.